OBSERVATION

# Overproduction of Chromosomal *ampC β*-Lactamase Gene Maintains Resistance to Cefazolin in *Escherichia coli* Isolates

Masato Kawamura,[a] Ryota Ito,[a] Yurina Tamura,[a] Mio Takahashi,[a] Miho Umenai,[b] Yuriko Chiba,[c] Takumi Sato,[a] Shigeru Fujimura[a]

[a]Division of Clinical Infectious Diseases & Chemotherapy, Tohoku Medical and Pharmaceutical University, Sendai, Japan
[b]Department of Pharmacy, Sendai Medical Center, Sendai, Japan
[c]Department of Pharmacy, Oosaki Citizen Hospital, Oosaki-shi, Japan

**ABSTRACT** Cefazolin, an active *in vitro* agent against *Escherichia coli*, is used to treat urinary and biliary tract infections. Cefazolin is used widely as an antibiotic, and the increase in the emergence of cefazolin-resistant *E. coli* in many countries is a major concern. We investigated the changes in the susceptibility of *E. coli* clinical isolates to cefazolin following exposure. A total of 88.9% (16/18 strains) of the strains acquired resistance to cefazolin. All strains with an MIC to cefazolin of 2 $\mu$g/mL became resistant. The expression of chromosomal *ampC* (c-*ampC*) increased up to 209.1-fold in the resistant strains. Moreover, 11 of the 16 *E. coli* strains (68.8%) that acquired cefazolin resistance maintained the resistant phenotype after subculture in cefazolin-free medium. Therefore, the acquisition and maintenance of cefazolin resistance in *E. coli* strains were associated with the overexpression of c-*ampC*. Mutations in the c-*ampC* attenuator regions are likely to be maintained and are one of the key factors contributing to the increase in the number of cefazolin-resistant *E. coli* worldwide.

**IMPORTANCE** This study is the first to demonstrate that mutations in the chromosomal-*amp*C attenuator region are responsible for the emergence of cefazolin resistance in *Escherichia coli* strains. The resistance was maintained even after culturing *E. coli* without cefazolin. This study highlights one of the key factors contributing to the increase in the number of cefazolin-resistant *E. coli* strains, which can pose a considerable challenge for treating common infections, such as urinary tract infections.

**KEYWORDS** cefazolin, *Escherichia coli*, acquired resistance, chromosomal-*ampC*

Cefazolin is a first-generation cephalosporin with bactericidal activity against *Escherichia coli*, *Streptococcus* spp., *Klebsiella* spp., and *Proteus mirabilis* in addition to staphylococci. It is used commonly as a prophylactic antibiotic for the prevention of surgical site infections according to the Centers for Disease Control and Prevention (1) and World Health Organization (WHO) guidelines (2). The WHO recommends cefazolin as a high-quality, inexpensive, and empirical first-line medication (3, 4). Therefore, cefazolin is used widely as an antibacterial drug in clinical settings for the treatment of infective endocarditis, joint infections, and skin infections caused by Gram-positive bacteria, such as methicillin-sensitive *Staphylococcus aureus*.

Antimicrobial de-escalation is recommended in the antimicrobial stewardship guidelines (5) to combat the emergence of antimicrobial resistance strains. Cefazolin is used commonly as a therapeutic antibiotic for infectious diseases caused by staphylococci; however, its use has been expanded to treat *E. coli* infections. The expansion has led to a worldwide increase in the number of cefazolin-resistant *E. coli* strains, reported at 12.1% to 34% in the United States (6, 7), 15.2% to 22.3% in Australia (8, 9), 63.6% in China (10), 39.5% in Taiwan (11), and 38.7% in Japan (12). *E. coli* is one of the most common causes of nosocomial and community-acquired bacterial infections, including urinary tract infections, enteric infections, and systemic infections along with more severe infections, such as bacteremia (13, 14). However, the precise mechanism underlying the increase in the number of cefazolin-resistant *E. coli*

Address correspondence to Masato Kawamura, m-kawamura@tohoku-mpu.ac.jp.

The authors declare no conflict of interest.

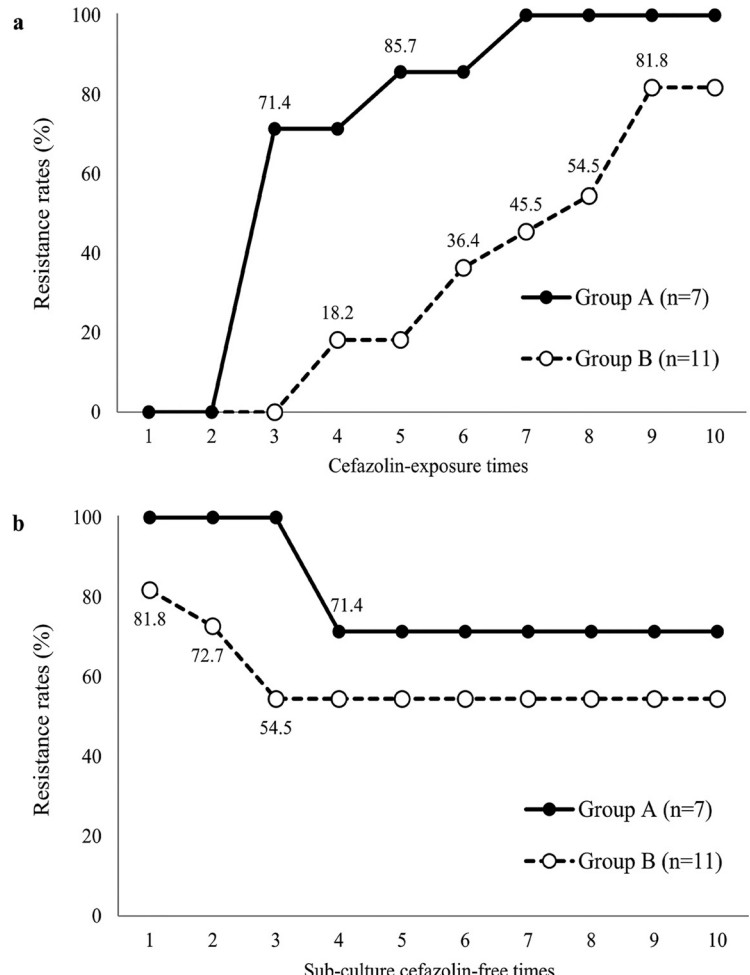

**FIG 1** Resistance rates in *Escherichia coli* isolates and the *E. coli* ATCC 25922 strain following culture in cefazolin-containing (a) and cefazolin-free medium (b). The MIC resistance breakpoints of cefazolin are ≥8 μg/mL, according to the Clinical and Laboratory Standards Institute guidelines. The cefazolin MICs in group A and B wild-type strains (i.e., prior to cefazolin exposure) were 2 μg/mL and 1 μg/mL, respectively.

strains remains unclear. A known antibacterial drug resistance mechanism is through the production of AmpC β-lactamases, encoded by *ampC* β-lactamase on chromosomes and/or plasmids; *E. coli* possesses a chromosomal *ampC* gene (c-*ampC*) (15). In this study, we investigated the mechanism underlying the acquisition and maintenance of resistance in *E. coli* following cefazolin exposure.

**Differences in the MIC between cefazolin-exposure and cefazolin-free subcultures.** Eighteen cefazolin-susceptible *E. coli* strains (MIC, ≤2 μg/mL) were selected from 43 non-duplicate clinical isolates. *E. coli* ATCC 25922 was used as the reference strain. Cefazolin MICs were determined using the broth microdilution method according to the Clinical and Laboratory Standards Institute guidelines and breakpoints (16).

Strains surviving at the sub-MIC were collected and then inoculated into a new 4 to 1/4 MIC cefazolin series in diluted Mueller-Hinton broth (MHB); this series was repeated 10 times. Among the 18 tested strains, 16 (88.9%) acquired resistance (MIC, ≥8 μg/mL) after 216 h of exposure. All *E. coli* clinical strains with an initial cefazolin MIC of 2 μg/mL (group A) acquired resistance, whereas 9 of the 11 strains (81.8%) with an initial cefazolin MIC of 1 μg/mL (group B) became resistant (Fig. 1a).

Among the 16 strains that acquired cefazolin resistance, 11 (68.8%) maintained the resistant phenotype after 10 repeated subcultures in cefazolin-free MHB. The resistance maintenance rates in groups A and B were 71.4% (5/7 strains) and 54.5% (6/11 strains),

respectively (Fig. 1b). Five strains (C-11, C-33, C-40, C-10, and C-12) showed particularly high resistance (MIC, $\geq$32 $\mu$g/mL) after repeated subculture in cefazolin-free medium (Table 1).

In this study, 41.2% of the cefazolin-sensitive strains had an MIC of 2 $\mu$g/mL, which is much higher than the 18% reported by Turnidge et al. in 2011 (17), suggesting that the MICs of cefazolin-sensitive *E. coli* are increasing. In this study, 71.4% strains became resistant within 3 days, highlighting the need to pay attention to *E. coli* cefazolin-sensitive strains with an MIC of 2 $\mu$g/mL.

**AmpC production.** To explore the link between AmpC production and acquisition of cefazolin resistance, real-time reverse transcription-quantitative PCR (RT-qPCR) was performed targeting the *ampC* gene. Total RNA was isolated from cefazolin-exposed and cefazolin-free subcultured *E. coli* strains using TRI Reagent LS (Molecular Research Center, Inc., Cincinnati, OH), according to the manufacturer's instructions. Expression of the c-*ampC* gene and the reference gene glyceraldehyde 3-phosphate dehydrogenase A (*gapA*) was assessed by RT-qPCR using the iTaq universal SYBR green one-step kit (Bio-Rad, CA). The relative expression of c-*ampC* mRNA was calculated as the fold change based on the mean normalized expression of c-*ampC* mRNA in the reference strain *E. coli* ATCC 25922 as 1.0. The following PCR primers were used: *ampC* forward primer 5'-TCAAACCAGACGGCTTC ACA-3' and reverse primer 5'-GTCTGTATGCCAACTCCAGTATCG-3', and *gapA* forward primer 5'-GGCCAGGACATCGTTTCCAA-3' and reverse primer 5'-TCGATGATGCCGAAGTTATCG TT-3' (18).

Following cefazolin exposure, the c-*ampC* mRNA expression was upregulated by 209.1-fold (Table 1), which is above the threshold of a 4.8-fold increase that indicates cefazolin resistance. Similarly, Paltansing et al. reported a 6.1- to 163.3-fold increase in c-*ampC* expression in clinical *E. coli* isolates that were resistant to cefoxitin and cefuroxime (19).

The c-*ampC* mRNA expression increased by 4.8- to 7.8-fold in group A strains that acquired resistance following the third cefazolin exposure; however, the strains with cefazolin MICs of $\geq$128 $\mu$g/mL showed a 134.7- to 209.1-fold increase in c-*ampC* mRNA expression. The exposure of *E. coli* to the sub-MIC of cefazolin for 3 days or longer led to increased c-*ampC* expression.

**Mutation detection in the AmpC promoter/attenuator regions.** Mutations in the c-*ampC* promoter/attenuator regions were detected via DNA sequencing at Eurofins Genomics K.K. using the primers AB1 5'-GATCGTTCTGCCGCTGTG-3' and ampC2 5'-GGGC AGCAAATGTGGAGCAA-3' (20). A total of 28 mutations were identified in the 18 isolates after cefazolin exposure, and at least 1 or more nucleotide changes were identified in each strain (Table 1). A total of 75% (12/16 strains) of the strains that acquired resistance had mutations at positions +17, +22, +24, +26, +27, +32, and +37 in the attenuator region (ranging from +17 to +37). Strains C-22 and C-40 (with cefazolin MICs of $\geq$128 $\mu$g/mL) had the following mutations: C→T at +22, T→G at +26, A→T at +27, and G→A at +32. Strain C-33 had the following mutations: C→T at +17, C→T at +22, G→A at +32, and G→A at +37. Furthermore, 81.8% (9/11) of the strains that maintained cefazolin resistance following subculture in the cefazolin-free condition harbored one to three mutations at positions +17, +22, +24, +32, and +37. The other two strains had the following mutations in the promoter regions: strain C-4 had C→T mutation at position −42 and strain C-27 had mutations T→C at −88, G→A at −82, and T→C at −1.

Mutations in the c-*ampC* attenuator regions reduce the transcriptional efficiency of RNA polymerase (15, 21), destabilize the stem-loop structure, and increase c-*ampC* gene transcription, resulting in the overproduction of AmpC $\beta$-lactamases (21–23). The mutations in the c-*ampC* attenuator region could be involved in maintaining cefazolin resistance in *E. coli*.

*E. coli* harboring extended-spectrum $\beta$-lactamase genes (9) and producing AmpC $\beta$-lactamase (21) are resistant to cefazolin. p-*ampC* genes, including CMY, ACC, ACT, FOX, MOX, and DHA, are involved in acquired antimicrobial resistance (18). p-*ampC* genes, such as CMY-2, are derived from *Citrobacter freundii* (24), whereas the *tet*(X) gene, involved in the tetracycline resistance in *E. coli*, originated from *Flavobacteriaceae* (25). Resistance genes, including *ampC* could be transmitted from other bacteria to *E. coli* through plasmids. In this study, the acquisition and maintenance of resistance were attributed to increased c-*ampC*

**TABLE 1** The minimum inhibitory concentration (MIC) of cefazolin, expression of chromosomal-*ampC* mRNA post cefazolin-exposure assay, and mutations in chromosomal-*ampC* promoter/attenuator region(s)

| Strain name by group | Cefazolin exposure[a] | | | | | | Wild-type strains exposed to cefazolin 10 times[b] | | Strains subcultured in cefazolin-free media post the cefazolin-exposure assay[b] | |
| --- | --- | --- | --- | --- | --- | --- | --- | --- | --- | --- |
| | Wild-type strain | | 3 times | | 10 times | | | | | |
| | MIC[c] | c-ampC expression[d] | MIC[c] | c-ampC expression[d] | MIC[c] | c-ampC expression[d] | MIC[c] | Position of mutation(s) | MIC[c] | Position of mutation(s) |
| **Group A[e]** | | | | | | | | | | |
| C-11 | 2 | 3.3 | 8 | 7.8 | 64 | 78.4 | 64 | +4/**+22/+32/+37**/+42/+44/+54/+65 | 32 | **+22/+32/+37**/+42/+65 |
| C-14 | 2 | 2.9 | 8 | 5.0 | 32 | 56.6 | 32 | -42/-1/**+17**/+65 | 16 | -42/**+17** |
| C-22 | 2 | 4.9 | 8 | 5.4 | 128 | 134.7 | 128 | -28/**+22/+26/+27/+32**/+54/+58 | 16 | -28/**+22/+32**/+58 |
| C-33 | 2 | 1.8 | 8 | 4.8 | >128 | 209.1 | >128 | -88/-82/-73/-42/-18/-8/**+17/+22/+32/+37**/+54 | 64 | -42/-18/-8/**+17/+22/+32/+37**/+54 |
| C-39 | 2 | 2.9 | 4 | 4.3 | 32 | 58.6 | 32 | -28/**+17**/+54/+58/+65 | 2 | -28/+65 |
| C-40 | 2 | 3.1 | 8 | 5.0 | >128 | 181.3 | >128 | -88/-82/-73/-42/-18/**+22/+26/+27/+32**/+54 | 32 | -42/-18/**+22/+32**/+54 |
| C-42 | 2 | 3.2 | 4 | 3.5 | 16 | 31.1 | 16 | **+32**/+63/+70/+80 | 4 | +63/+70/+80 |
| **Group B[f]** | | | | | | | | | | |
| C-1 | 1 | 0.4 | 2 | 1.9 | 4 | 6.9 | 4 | **+24** | 2 | ND[g] |
| C-4 | 1 | 2.1 | 4 | 2.8 | 8 | 11.7 | 8 | -76/-42/-1 | 8 | -76/-42/-1 |
| C-5 | 1 | 1.1 | 4 | 1.7 | 8 | 6.4 | 8 | **+24**/+54 | 8 | **+24** |
| C-10 | 1 | 0.8 | 2 | 1.1 | 32 | 41.6 | 32 | -73/-28/**+17/+24** | 32 | -73/-28/**+24** |
| C-12 | 1 | 0.4 | 2 | 1.5 | 32 | 34.7 | 32 | -73/**+22/+32** | 32 | -73/**+22/+32** |
| C-13 | 1 | 1.9 | 2 | 1.5 | 8 | 6.2 | 8 | -28/**+17**/+54/+81 | 4 | **+17**/+81 |
| C-21 | 1 | 0.4 | 2 | 1.9 | 8 | 7.0 | 8 | -42/+6/+50/+54 | 4 | -42/+50 |
| C-27 | 1 | 0.5 | 2 | 1.3 | 16 | 12.8 | 16 | -88/-82/-1/+50/+81 | 8 | -88/-82/-28/-1/+50/+81 |
| C-35 | 1 | 0.2 | 2 | 1.2 | 4 | 4.2 | 4 | -73/**+37**/+63 | 4 | -73/-28/**+37** |
| C-37 | 1 | 0.5 | 4 | 1.8 | 16 | 18.9 | 16 | -28/-1/**+24** | 8 | -28/**+24** |
| *E. coli* ATCC 25922 | 1 | 1.0 | 2 | 1.5 | 8 | 6.8 | 8 | +50/+63/+81 | 4 | +63/+81 |

[a] The relative expression of chromosomal-*ampC* mRNA in wild-type *E. coli* ATCC 25922 was set to 1.0.
[b] Mutations in the chromosomal-*ampC* attenuator region are indicated in boldface.
[c] Numbers represent concentration in μg/mL.
[d] Values represent fold change.
[e] Cefazolin MIC for the wild-type strain is 2 μg/mL.
[f] Cefazolin MIC for the wild-type strain is 1 μg/mL.
[g] ND, not detected.

production. The clinical isolates harbored the *c-ampC* gene; therefore, it is necessary to restrict cefazolin use.

In summary, 2 days of cefazolin administration should be sufficient in cases of cefazolin-susceptible *E. coli* infections. Subsequent cefazolin administration should be assessed using routine monitoring of cefazolin MIC values.

## ACKNOWLEDGMENT

This research received no specific grant from any funding agency in the public, commercial, or not-for-profit sectors.

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
