## [Reviewer comments · Microbiology Spectrum]

Microbiology Spectrum

Overproduction of chromosomal *ampC* β -lactamase gene maintains resistance to cefazolin in *Escherichia coli* isolates

Masato Kawamura, Ryota ITO, Yurina TAMURA, Mio TAKAHASHI, Miho Umenai, Yuriko Chiba, Takumi Sato, and Shigeru Fujimura

Corresponding Author(s): Masato Kawamura, Tohoku Medical and Pharmaceutical University

Review Timeline:

Submission Date:	January 13, 2022
Editorial Decision:	February 27, 2022
Revision Received:	March 24, 2022
Editorial Decision:	April 5, 2022
Revision Received:	April 25, 2022
Accepted:	May 8, 2022

Editor: Hui Wang

Reviewer(s): The reviewers have opted to remain anonymous.

Transaction Report:

DOI: <https://doi.org/10.1128/spectrum.00058-22>

February 27, 2022

Dr. Masato Kawamura
Tohoku Medical and Pharmaceutical University
Division of Clinical Infectious Diseases & Chemotherapy
4-4-1 Komatsushima
Aoba-ku
Sendai, Miyagi-ken 981-8558
Japan

Re: Spectrum00058-22 (Overproduction of chromosomal *ampC* β -lactamase gene maintains resistance to cefazolin in *Escherichia coli* isolates)

Dear Dr. Masato Kawamura:

Thank you for submitting your manuscript to Microbiology Spectrum. The MS lacks of originality as AAC editor indicated. Moreover, the sample is limited. Please make the major revision. When submitting the revised version of your paper, please provide (1) point-by-point responses to the issues raised by the reviewers as file type "Response to Reviewers," not in your cover letter, and (2) a PDF file that indicates the changes from the original submission (by highlighting or underlining the changes) as file type "Marked Up Manuscript - For Review Only". Please use this link to submit your revised manuscript - we strongly recommend that you submit your paper within the next 60 days or reach out to me. Detailed instructions on submitting your revised paper are below.

Link Not Available

Sincerely,

Hui Wang

Journals Department
Reviewer comments:

Reviewer #1 (Comments for the Author):

Cefazolin is an active treatment of urinary tract infections and biliary tract infections. Whereas a concern is raising towards an increase in the emergence of cefazolin-resistant *E. coli*.

Authors reported acquired resistance to cefazolin in certain E. coli strains, and determined the underlying mechanism is due to the expression of chromosomal ampC. We are very curious if plasmid-borne-ampC is present or not!? This is a big problem for public health! Authors are requested to integrate such information or discuss it. Similar scenarios were seen with 1) chromosomal EptA/ plasmid Mcr-1 (1 to 10) (Zhang et al., Trends Biochem Sci, 2019); 2) chromosomal TetX/ plasmid-borne Tet(X) (WIREs Mech Dis, 2022).

Also, the language errors are accumulated, and recommended to revise accordingly.

Reviewer #2 (Comments for the Author):

Generally, the authors describe an interesting finding of the mechanism of cefazolin resistance in E. coli strains. However, the referee suffers from the poor presentation of results to go over the manuscript. Besides, there are still some questions needed to be interpreted in details to make the rigor of the manuscript.

One of the major comment is the that sample scale is too small, only 18 strains. Furthermore, is the observation specific to cefazolin? What about other beta-lactams? Why?

My detailed comments are as follows:

1. To present the results of the study more clearly, it would be better to divide the text into different parts as Introduction, Materials and methods, Results and Discussion.
2. Line 31: Specific number should be given, as "61%" is not clear enough.
3. Line 97: "Mueller-Hinton (MHB) broth" should be modified as "Mueller-Hinton broth (MHB)".
4. Line 99: Please change "day 9" to the corresponding passage times.
5. Line 100, 104, 106: Data should be shown in the corresponding figures.
6. Line 102-103: Purpose of presenting this result should be illustrated.

Staff Comments:

Preparing Revision Guidelines

Please return the manuscript within 60 days; if you cannot complete the modification within this time period, please contact me. If you do not wish to modify the manuscript and prefer to submit it to another journal, please notify me of your decision immediately so that the manuscript may be formally withdrawn from consideration by Microbiology Spectrum.

March 24, 2022

RESPONSE TO REVIEWER #1

We wish to express our appreciation to Reviewer #1 for their insightful comments on our paper. In particular, we wish to acknowledge the Reviewer's highly valuable comments regarding the presence of plasmid-borne-*ampC*.

Reviewer comment 1: Authors reported acquired resistance to cefazolin in certain *E. coli* strains, and determined the underlying mechanism is due to the expression of chromosomal *ampC*. We are very curious if plasmid-borne-*ampC* is present or not!? This is a big problem for public health! Authors are requested to integrate such information or discuss it. Similar scenarios were seen with 1) chromosomal EptA/ plasmid Mcr-1 (1 to 10) (Zhang et al., Trends Biochem Sci, 2019); 2) chromosomal TetX/ plasmid-borne Tet(X) (WIREs Mech Dis, 2022).

Response: We appreciate your comment on this topic. A multiplex PCR was performed for the six families of *p-ampC* genes (J Clin Microb, 40, 2002). We have added the following sentence at lines 141–142 (underlined):

“However, none of our tested strains harbored *p-ampC* genes (ACC, CIT, DHA, EBC, FOX, and MOX) (J Clin Microb, 40, 2002) (data not shown).”

Reviewer comment 2: Also, the language errors are accumulated, and recommended to revise accordingly.

Response: We have proofread the English text prior to resubmission.

We have worked to incorporate your feedback and hope that the revised manuscript is now suitable for publication in your journal. We look forward to hearing from you and responding to any further questions and comments that you may have.

Yours sincerely,

Masato KAWAMURA Ph.D.

Tohoku Medical and Pharmaceutical University

4-4-1 Komatsushima, Aoba-ku, Sendai-shi 981-8558, Japan

Phone: +81-22-727-0176 Fax: +81-22-391-8525

Email: m-kawamura@tohoku-mpu.ac.jp

March 24, 2022

RESPONSE TO REVIEWER #2

Submission of revised paper #Spectrum00058-22R1 titled “Overproduction of Chromosomal *ampC* β -Lactamase Gene Maintains Resistance to Cefazolin in *Escherichia coli* Isolates.”

We wish to express our appreciation to Reviewer #2 for their insightful comments on our paper. In particular, we wish to acknowledge the Reviewer’s highly valuable comments regarding the mechanism of maintained cefazolin resistance in *E. coli* strains.

Reviewer comment: One of the major comment is the that sample scale is too small, only 18 strains. Furthermore, is the observation specific to cefazolin? What about other beta-lactams? Why?

Response: We appreciate the Reviewer’s comment on this topic. A total of 44 clinical isolates of *E. coli*, including *E. coli* ATCC 25922, were preserved in our laboratory. We selected 18 (40.9%) of the 44 strains that were resistant to cefazolin. The proportion of cefazolin-resistant *E. coli* has been reported as “12.1–34% in the United States, 15.2–22.3% in Australia, 63.6% in China, 39.5% in Taiwan, and 38.7% in Japan” (lines 76–78). Thus, based on the proportion, we believe it is sufficient to conclude that the overproduction of the chromosomal *ampC* gene maintains cefazolin resistance in *E. coli* isolates.

Cefazolin is often used clinically because of its antibacterial activity against *E. coli*. In addition, the isolation rate of resistant strains is increasing. Therefore, this study focused on revealing the mechanism of maintained cefazolin resistance.

Reviewer detailed comment 1. To present the results of the study more clearly, it would be better to divide the text into different parts as Introduction, Materials and methods, Results and Discussion.

Response: Thank you for your comments. We have written “Introduction”, “Materials and methods, Results and Discussion”, as per your suggestion.

Reviewer detailed comment 2. Line 31: Specific number should be given, as "61%" is not clear enough.

Response: We have revised the following sentence at lines 30–32.

“Moreover, 11 of the 16 *E. coli* strains (68.8%) that acquired cefazolin resistance maintained this resistant phenotype after subculture in cefazolin-free medium.”

Reviewer detailed comment 3. Line 97: "Mueller-Hinton (MHB) broth" should be modified as "Mueller-Hinton broth (MHB)".

Response: We have revised as "Mueller-Hinton broth (MHB)" (line 97).

Reviewer detailed comment 4. Line 99: Please change "day 9" to the corresponding passage times.

Response: We changed "day 9" to "216 h" (line 99) as per your suggestion.

Reviewer detailed comment 5: Line 100, 104, 106: Data should be shown in the corresponding figures.

Response: As suggested, we have expressed percentages in Figure 1. We have revised the following sentence at lines 103–105 according to Figure 1.

"The resistance maintenance rates in groups A and B were 71.4% (5/7 strains) and 54.5% (6/11 strains), respectively (Fig. 1-b)."

Reviewer detailed comment 6:

Line 102-103: Purpose of presenting this result should be illustrated.

Response: Thank you for your comments. We removed the sentence at lines 102–103 because the same content was explained at lines 110–112.

We have worked to incorporate your feedback and hope that the revised manuscript is now acceptable for publication in your journal. We look forward to hearing from you and responding to any further questions and comments that you may have.

Yours sincerely,

Masato KAWAMURA Ph.D.

Tohoku Medical and Pharmaceutical University

4-4-1 Komatsushima, Aoba-ku, Sendai-shi 981-8558, Japan

Phone: +81-22-727-0176

Fax: +81-22-391-8525

Email: m-kawamura@tohoku-mpu.ac.jp

April 5, 2022

Dr. Masato Kawamura
Tohoku Medical and Pharmaceutical University
Division of Clinical Infectious Diseases & Chemotherapy
4-4-1 Komatsushima
Aoba-ku
Sendai, Miyagi-ken 981-8558
Japan

Re: Spectrum00058-22R1 (Overproduction of chromosomal *ampC* β -lactamase gene maintains resistance to cefazolin in *Escherichia coli* isolates)

Dear Dr. Masato Kawamura:

Link Not Available

Sincerely,

Hui Wang

Journals Department
Reviewer comments:

Reviewer #1 (Comments for the Author):

The response (e.g.,: one sentence for comment 1) cannot satisfy the minimal requirement of this journal. Also, the authors fail to integrate all the relative literatures to discuss the potential significance and threat of such plasmid-borne resistance in the context of one health. Authors are suggested to compare different resistance mechanisms, and extend the possible common mechanism for transmission and action.

Staff Comments:

Preparing Revision Guidelines

Please return the manuscript within 60 days; if you cannot complete the modification within this time period, please contact me. If you do not wish to modify the manuscript and prefer to submit it to another journal, please notify me of your decision immediately so that the manuscript may be formally withdrawn from consideration by Microbiology Spectrum.

April 25, 2022

RESPONSE TO REVIEWER #1

Submission of the revised paper titled, “Overproduction of Chromosomal *ampC* β -Lactamase Gene Maintains Resistance to Cefazolin in *Escherichia coli* Isolates,” manuscript ID: #Spectrum00058-22R2.

We wish to express our appreciation to Reviewer #1 for the insightful comments on our paper. In particular, we wish to acknowledge the constructive suggestions, which have helped us improve the paper considerably.

Reviewer comment: The response (e.g.,: one sentence for comment 1) cannot satisfy the minimal requirement of this journal. Also, the authors fail to integrate all the relative literatures to discuss the potential significance and threat of such plasmid-borne resistance in the context of one health. Authors are suggested to compare different resistance mechanisms, and extend the possible common mechanism for transmission and action.

Response: Thank you for your comments. In response, we have included the following information in the Discussion at lines 163–171.

Line 163–171: “*E. coli* harboring extended-spectrum β -lactamase genes and producing AmpC β -lactamase are resistant to cefazolin. *p-ampC* genes including CMY, ACC, ACT, FOX, MOX, and DHA are involved in acquired antimicrobial resistance. *p-ampC* genes such as CMY-2 are derived from *Citrobacter freundii*, whereas the *tet(X)* gene, involved in the tetracycline resistance in *E. coli*, originated from Flavobacteriaceae. Resistance genes including *ampC* could be transmitted from other bacteria to *E. coli* through plasmids. In this study, the acquisition and maintenance of resistance was attributed to increased *c-ampC* production. The clinical isolates harbored the *c-ampC* gene; therefore, it is necessary to restrict cefazolin use.”

We have worked diligently to incorporate changes according to feedback received and hope that the revised manuscript is now suitable for publication in *Microbiology spectrum*. We look forward to hearing from you and responding to any further questions and comments that you may have.

Yours sincerely,

Masato KAWAMURA Ph.D.

Tohoku Medical and Pharmaceutical University

4-4-1 Komatsushima, Aoba-ku, Sendai-shi 981-8558, Japan

Phone: +81-22-727-0176 Fax: +81-22-391-8525

Email: m-kawamura@tohoku-mpu.ac.jp

May 8, 2022

Dr. Masato Kawamura
Tohoku Medical and Pharmaceutical University
Division of Clinical Infectious Diseases & Chemotherapy
4-4-1 Komatsushima
Aoba-ku
Sendai, Miyagi-ken 981-8558
Japan

Re: Spectrum00058-22R2 (Overproduction of chromosomal *ampC* β -lactamase gene maintains resistance to cefazolin in *Escherichia coli* isolates)

Dear Dr. Masato Kawamura:

Your manuscript has been accepted, and I am forwarding it to the ASM Journals Department for publication. You will be notified when your proofs are ready to be viewed.

Sincerely,

Hui Wang
Editor, Microbiology Spectrum
